# Doing Work on the Land of Our Ancestors: Reserved Treaty Rights Lands Collaborations in the American Southwest

**Gregory Russell** [1,*], **Joseph G. Champ** [1], **David Flores** [2], **Michael Martinez** [3], **Alan M. Hatch** [4], **Esther Morgan** [5] **and Paul Clarke** [6]

1   Department of Journalism and Media Communication, Colorado State University, Fort Collins, CO 80523, USA; joseph.champ@colostate.edu
2   Rocky Mountain Research Station, USDA Forest Service, Fort Collins, CO 80526, USA; david.flores2@usda.gov
3   Department of Environment and Natural Resources, Pueblo of Tesuque, NM 87506, USA; michaelm@pueblooftesuque.org
4   Department of Natural Resources, Pueblo of Santa Ana, NM 87004, USA; alan.hatch@santaana-nsn.gov
5   Apache-Sitgreaves National Forests Supervisor's Office, Springerville, AZ 85938, USA; esther.morgan@usda.gov
6   Department of Natural Resources, Jemez Pueblo, NM 87024, USA; paul.clarke@jemezpueblo.org
*  Correspondence: gregory.russell@colostate.edu

**Abstract:** The intent of this article is to raise awareness about an underutilized funding mechanism that possesses the capacity to help tribal and federal land management agencies meet their goal of restoring fire-adapted ecosystems to historic conditions in the American Southwest. We attempt to achieve this through an exploration of the Reserved Treaty Rights Lands (RTRL) program and how it has been used to implement collaborative fuel management projects on National Forest lands. RTRL is a funding program administered by the Bureau of Indian Affairs (BIA) that is designed to protect natural and cultural resources important to tribes on non-tribal lands that are at high risk from wildfire. Over the last year, our research team has studied the RTRL program in the Southwest by conducting in-depth, face-to-face interviews with tribal land managers as well as U.S. Forest Service tribal liaisons and other personnel who work with tribes. Our interviews revealed enthusiasm and support for RTRL but also concern about the fairness of the program as well as insufficient outreach efforts by the U.S. Forest Service. In response, we propose a policy alteration that (we contend) would incentivize the BIA to increase funding allocations to the RTRL program without losing the support of partnering agencies. The aim is to strengthen and expand shared stewardship efforts between tribes and federal land management agencies. We situate these implications against the backdrop of the Pacheco Canyon Prescribed Burn, an RTRL funded project that was instrumental in containing the Medio Fire that broke out in the Santa Fe National Forest in the summer of 2020.

**Keywords:** fuel treatment; prescribed burning; tribal collaboration; fire management

## 1. Introduction

As wildfires increase in size and intensity throughout the Southwest region of the United States [1], scientists and land managers are searching for ways to fund fuel-treatment projects to responsibly address the accumulation of fuel loads that have amassed on forest floors [2]. Despite widespread agreement about the benefits of prescribed burning to mitigate the scale and intensity of wildfire, the current rate of treatments does not adequately address previous decades of wildfire suppression [3], including the high hazard areas of the American West and Southwest [4].

In this paper, we consider the Reserved Treaty Rights Lands (RTRL) program—a funding mechanism administered by the Bureau of Indian Affairs (BIA)—and how it is being utilized in the American Southwest to mitigate the risk of high intensity wildfires. After sketching a recent example of a successful fuel treatment project using RTRL funds,

we explore some of the program's limitations as described by several key actors in the Southwest responsible for the implementation of the RTRL program. We close the paper by proposing a policy alteration to RTRL to address some of these limitations. Our intent is to bring awareness of an underutilized funding mechanism that possesses the capacity to help federal and tribal land-management agencies meet their goal of restoring fire adapted ecosystems to historic conditions and to strengthen shared stewardship efforts between tribes and federal land management agencies.

## 2. Background and Methods

A primary objective of prescribed burning is to safely return fire back to fire-dependent ecosystems in a manner that is consistent with historic conditions [5]. In the Southwest (as well as many other locations throughout North America), historic conditions were, in part, determined by the cultural application of fire to forest lands by indigenous peoples. As Pyne notes, "The fire regime of the Southwest has been shaped by lightning and livestock, but the Apache was for centuries the intervening variable" [6].

When the original inhabitants of what is now referred to as the Americas were removed from their ancestral lands by colonial forces, so too was an ecological balance that had sustained people and the land for thousands of years [7]. In North America, the westward expansion of Euro-American settlement brought with it a conviction that wildfire should be suppressed. This view ran counter to the traditional burn practices of many Native American tribes, who had long used fire as a land management tool [8].

Through a comprehensive review of over 300 studies, Williams [9] identified 11 categories of traditional fire used by indigenous Americans: hunting, managing crops, improving the growth and yield of wild plants, fireproofing areas around settlements, collecting insects, managing pests, waging war, extorting trade benefits from settlers and trappers by depriving them of easy access to big game (scorched earth policy), clearing travel routes, felling trees, and clearing riparian areas. Despite the crucial historical role these practices played in modulating forest ecologies, early generations of U.S. Forest Service land managers, motivated by an ethos of profit and sustainability, regarded wildfire suppression as a vehicle capable of delivering an abundant and verdant future to the United States [10]. This conviction was formally instituted in the Forest Service through the adoption of an agency-wide fire suppression program that lasted until the early 1970s [11].

However, it is now widely recognized by most forest and fire ecologists that the wildfire suppression strategies of the 20th century have contributed significantly to functional and structural changes in forest ecologies throughout the Western United States [12–14]. Along with increased drought and temperatures [15,16], the buildup of fuel loads from wildfire suppression is considered a primary contributing factor to the rise in high intensity forest fires [17]. The shift in fire regime from frequent low-intensity burns to suppression has provided indigenous land practitioners with a new use for fire: to restore fire-dependent ecosystems to precolonial conditions.

Over the last year, our research team has studied collaborative partnerships between the U.S. Forest Service and tribes in the Southwest. On two separate occasions (from fall 2019 to winter 2020), the first author traveled to New Mexico and Arizona to conduct semistructured in-depth interviews with tribal land managers as well as U.S. Forest Service personnel who work with tribes. An interview can be understood as "semi-structured" when it has a preplanned set of general questions that the researcher asks all participants but also gives the researcher freedom to vary the questions as the situation demands [18]. An interview can be understood as "in-depth" when the researcher seeks to achieve the same deep level of knowledge and understanding about a topic(s) as the participants [19].

Additionally, this study attempted to ground itself in a participatory research orientation. Participatory research is considered an alternative approach to traditional social science, as it repositions social investigation from a linear cause and effect orientation to a collaborative approach focused on the contexts of people's lives [20,21]. The intent is to design and conduct research studies that promote self-determination, a key component

that has "been missing from much research involving indigenous communities in the past" [22]. By making space for the interpretations of participants in research outcomes, participatory research provides community members a chance to author their own stories instead of being spoken for by researchers [23]. Participatory research, then, can be viewed as a response to a history of Western researchers' extracting and claiming ownership over indigenous ways of knowing while at the same time rejecting "the people who developed those ideas" by denying "them further opportunities to be creators of their own culture and own nations" [24].

As such, participants in this study were given the opportunity to review and, ultimately, approve how they are represented in this report. Each participant represented in the following pages was sent a document with excerpts of every instance where he or she is mentioned in the analysis. If a participant agreed with the way he or she was represented, we kept it as is. If a participant was dissatisfied (for whatever reason), we worked together to find a way to represent the participant in a manner that was acceptable to him or her. This research orientation positions us to understand participants as knowledgeable and socially aware beings who have the capacity to make sense of their own social lives [25,26] and that researchers have the capacity to describe these perspectives accurately and with credibility [27].

## 3. Case Study: The Pacheco Canyon Prescribed Burn

On 17 August 2020, lightning sparked the Medio Fire in the Santa Fe National Forest, just north of Santa Fe, New Mexico [28]. For 28 days, the wildfire burned over 4000 acres [29], with 30 percent of these acres burning at moderate to high severity [30].

Firefighters confined and eventually extinguished the fire by guiding it to an existing burn scar from a 2011 wildfire, which limited its progress to the north. To the south, firefighters herded the flames to an area of forest that had been treated with a prescribed burn in 2019 [31]. U.S. Forest Service officials credit this prescribed burn—known as the Pacheco Canyon Prescribed Burn—with creating a fuel break that prevented the Medio Fire from gaining intensity, consuming additional acres of forest, degrading Santa Fe's watershed, and spreading to a local ski area [32].

The Pacheco Canyon Prescribed Burn was financed through the RTRL program. According to the Department of Interior's (DoI's) *Budget Justification for 2020*, "the RTRL program upholds our trust responsibilities by supporting Tribes' participation in collaborative strategic fuels management projects on non-Tribal lands to protect priority Tribal natural resources that are at high risk from wildfire" [33], though (as several participants in our study pointed out), it is possible to do RTRL fuel treatment work on Tribal lands, so long as it adds to the aim of the overall project.

In 2019, after securing RTRL funds from the BIA, the Pueblo of Tesuque partnered with the U.S. Forest Service (and other regional agencies) to initiate a 500-acre prescribed burn, within the 2400-acre Pacheco Canyon Project. Despite Tesuque lands not receiving any direct fuel treatments from the prescribed burn (though it should be noted that Tesuque lands did receive fuel buffers as part of the broader Pacheco Canyon Project), the Pueblo of Tesuque's Department of Environment and Natural Resources gained access to new resources and revenue streams that allowed the department to expand its capacity. The RTRL Coordinator for the Pueblo of Tesuque explained that, with RTRL funds, the Department's Wildfire Response Unit participated and received training for their Incident Qualification Cards.

However, Tesuque acquired more than training from the Pacheco Canyon Prescribed Burn. They also gained a fair and advantageous partnership with the U.S. Forest Service. Tesuque's RTRL Coordinator described the partnership as a "50/50 collaboration". The equitable nature of this collaboration registered during the National Environmental Protection Act (NEPA) review process, which is required for all prescribed fire projects on federal lands [34]. NEPA mandates assessments of both cultural and environmental impacts before any fuel treatment operation can begin [35]. For the Pacheco Canyon Prescribed

Burn, Tesuque administered the cultural survey while the U.S. Forest Service managed the environmental assessment.

In our interviews, it was common for participants from the US Forest Service to stress the importance of letting tribes conduct cultural surveys on federal lands, especially on lands that are part of a tribe's ancestral domain. For example, the Forest Archaeologist for the Apache-Sitgreaves National Forest expressed an appreciation for the long history tribes have with the landscape, arguing that the original inhabitants of the land "should be number one on the list of collaborators to help with restoration projects." The traditional knowledge that tribal elders bring to archaeological projects has augmented her formal training in beneficial ways, and she provided several examples of learning from tribal elders in the field that led to new discoveries and approaches to looking at landscapes.

Collins et al. note that landscape-level fuel treatment projects can take several years to complete and that "forest managers are often limited in time, and in some cases expertise, when conducting these comprehensive evaluations" [36]. Allowing tribes to conduct cultural surveys on federal lands for fuel treatment projects induces a level of cultural expertise that goes beyond what can be learned in a classroom or textbook. As the RTRL Coordinator for the Pueblo of Tesuque put it, "Although we implement projects on Forest Service lands, we're still doing work on the land of our ancestors, our aboriginal lands."

## 4. Some Limitations and a Policy Alteration

With the success of the Pacheco Canyon Burn (and other RTRL fuel treatment projects), it is clear that the RTRL program possesses the collaborative and financial capacity to assist tribal and federal land management agencies with addressing the glut of fuel loads that have accumulated on the forest floors of the Southwest. What remains less clear is why allocations for the RTRL program are underrepresented in the DoI's budget, accounting for just 0.05 percent of DoI's 2020 fuel treatment program [33].

A Partnership Coordinator for the Southwestern Region of the U.S. Forest Service provided a possible reason why the DoI does not put more financial support behind the RTRL program: The BIA does not want tribal fire crews who have been trained with BIA monies to be doing work on non-tribal lands, since much of BIA's funding is tied to project accomplishments that occur on tribal territories. Tribes do not get project accomplishments for doing work on U.S. Forest Service lands, so the US Forest Service gets all of the accomplishments for RTRL projects. He wants to see the US Forest Service and BIA figure out a way to adjust project accomplishments and reports so that BIA funding opportunities are not dinged for providing RTRL-trained crews to do work off reservation lands.

Even though most tribal land managers praised the success of RTRL fuel treatment projects in the Southwest, others expressed frustration that the program limits projects—and the accomplishments that follow—to non-tribal lands. The Natural Resource Director for the Pueblo of Santa Ana explained that there is already enough of a need on tribal trust lands that tribes should be sufficiently funded by the DoI to meet its trust responsibilities. He went on to state that the way RTRL is set up is unfortunate: federal agencies have thousands of employees, whereas the Santa Ana Natural Resource Department only has a 4-person crew with 140,000 acres to manage. "We have enough need to do work on our lands." The federal government has a fiduciary responsibility to protect trust lands, and this is why the RTRL program could go to funding work on tribal lands as well.

Other tribal land managers voiced concern about the lack of outreach efforts by the U.S. Forest Service in pursuing RTRL partnerships with regional tribes. A Natural Resources Director for one of the Pueblos in our study explained that his department has tried to "get some traction" with the U.S. Forest Service to start an RTRL project, but "we just haven't had a lot of interactions with the Forest Service's RTRL folks." The Director reported that when he first spoke with U.S. Forest Service representatives about the RTRL program, they did not make it clear who he would have to speak with about initiating a project.

In light of these limitations, we want to suggest an alteration to the RTRL program that would not only incentivize the BIA to increase the RTRL funding slice in its suppression

budget but also enhance the intercultural competence and outreach capacities of federal land management agencies. Obviously, we are not policy experts privy to the elaborate power networks that ultimately determine how and why monies get allocated at the BIA; however, it seems plausible that if the BIA is capable of funding tribes and their federal partners to implement extensive fuel treatment projects on non-tribal lands, they could likely do the same on lands that *do* belong to tribes. What we are advancing is the possibility of proportional representation of treated acres between tribes and their partnering agencies for RTRL projects. Simply put: for approximately every acre treated with RTRL funds on federal lands, there would be an equal amount treated on tribal lands.

As noted above (in the previous section), some of the participants we interviewed explained that it is already possible for RTRL projects to also treat acreage on adjacent tribal lands. However, the fuel treatments that occur within tribal boundaries are subsidiary to the *primary* fuel treatment project that occurs beyond tribal domains. This means that the total number of acres treated on tribal lands is typically much less than those treated on the federal side, even though—as we were told numerous times by multiple tribal land managers—there is just as much of a need for fuel treatment and restoration work on tribal lands as there is on federal lands.

We maintain that altering the RTRL program so that fuel treatment efforts proportionally benefit tribal and federal lands would lead to two positive outcomes:

1. The BIA would receive project accomplishments for fuel treatments completed on tribal lands. From a basic return-on-investment standpoint, it does not make sense for the BIA to allocate substantial monies from their suppression budget to projects that primarily benefit lands that fall under the territorial domains of other federal agencies. Treating more acres of tribal land through RTRL monies would give the BIA more justification to fund RTRL projects since these fuel treatment projects would take place directly on BIA administered or supervised lands.

2. From the same return-on-investment standpoint, shifting RTRL project funds from federal to tribal lands would disincentivize the U.S. Forest Service and other federal land management agencies from participating in the RTRL program as tribal partners, since less acres would be treated on their side of the fence. However, what the U.S. Forest Service would lose in acres treated, it would gain in its capacity to develop as an *intercultural and participatory management* agency, which Eloy et al. [37] define as the "equitable participation of different stakeholders in the process of planning and decision making based on the promotion of respect and mutual understanding among stakeholders, with different knowledge, needs and worldviews."

To understand why the U.S. Forest Service might be willing to sacrifice acres treated for intercultural growth in its management practices, it is necessary to understand the extent to which it takes seriously its relations with tribal communities. According to the most recent edition of its *Tribal Relations Strategic Plan* [38], the U.S. Forest Service acknowledges that tribes possess "indelible ties to the Nation's forests and grasslands" and that these ties affect "current knowledge, perspectives, and resources that will help the Forest Service as we focus on the future of our mission." The same report goes on to state that the U.S. Forest Service is committed to seeking "opportunities to partner with Tribes in work across boundaries and leverage resources to accomplish together what we could not each do on our own."

Based on these statements, it seems evident that the U.S. Forest Service values the knowledge of its tribal partners, to such an extent, in fact, that the agency is open to letting tribal knowledge influence its organizational aims. Moreover, the U.S. Forest Service not only appears interested in developing partnerships with tribal communities but in pursuing partnerships that take place "across boundaries".

As such, the basis for intercultural and participatory management already exists in the institutional discourse of the U.S. Forest Service, and in many ways, it has already taken active steps toward meeting this ideal, as evidenced by the agency's willingness to partner with tribes on a variety of projects, including through the RTRL program. However, most

of the partnerships reported in our interviews have taken place on National Forests. The few collaborative projects that we did hear about taking place on tribal lands were small in scale and infrequent.

Allowing RTRL projects to extend onto tribal lands would give tribal officials the final say over how their portion of the project is designed and implemented. Even though tribal land management departments would still be required to comply with NEPA regulations (since RTRL is financed with federal funds), the fact that the U.S. Congress has recognized tribes as sovereign governments gives them greater autonomy over the development and implementation of federal programs, per the Tribal Self-Governance Act of 1994 [39].

Accordingly, it is quite possible that tribal land managers would design and implement their portion of an RTRL fuel treatment project distinctly from their federal peers. A project orientation such as this would present U.S. Forest Service land managers with a unique learning opportunity, one where they would not only be in a position to cull knowledge from their tribal peers but also assist in the application of that knowledge, moving the agency closer toward a management ideal rooted in intercultural competence and participatory action.

## 5. Final Thoughts

Even though there is room for improvement in the RTRL program, the success of recent fuel treatment partnerships has shown that RTRL funds and projects can be used to cultivate stronger relationships that develop mutual benefits:

For example, the success of the Pacheco Canyon Prescribed Burn (as well as other collaborations) illustrates that interagency fuel treatment projects are capable of restoring the health and resiliency of fire adapted ecosystems. These outcomes show that when federal agencies collaborate with tribes on fuel treatment projects, they gain qualified partners who can manage forest lands on par with federal agencies. A Partnership Coordinator for the Southwestern Region of the U.S. Forest Service recognized that the RTRL program has historically been a hard sell for agency leadership because it requires them to relinquish power to tribes, but he remains optimistic about the future of the RTRL program, as a new generation of leaders who embrace shared stewardship has taken root in the agency.

In addition, putting fewer resources into fuel treatment projects allows federal land management agencies to maintain their commitment to restoring fire adapted ecosystems to historic conditions [40] while saving significant financial and personnel resources that can be directed toward other pressing circumstances, such as wildfire suppression efforts. When the US Forest Service partners with tribes on RTRL projects, they are securing access to an external source of funding (via the BIA) that effectively circumvents the suppression-focused budget directives that currently constrain fuel treatment efforts in the agency [41]. The Natural Resource Director for Jemez Pueblo acknowledged that RTRL makes it easier to partner with federal agencies, since his department is able to bring in money for projects federal partners want done.

The benefits of the RTRL program (even in its current configuration) do not end with federal land management agencies. As a grant coordinator and forestry worker for a regional pueblo explained, the RTRL program is a means of keeping their forestry department operational while simultaneously providing them with an "opportunity to get involved in projects that protect the lands that we have left." From this vantage point, the RTRL program can be viewed as a funding vehicle capable of moving the federal government towards meeting its funding obligations to tribal communities, which have been chronically and significantly unmet for decades [42].

Our intent with this paper is to bring awareness about as well as enhance (potentially) an underutilized funding mechanism that possesses the capacity to help federal land management agencies meet their goal of restoring fire adapted ecosystems to historic conditions and to strengthen shared stewardship efforts between tribes and federal land management agencies. The RTRL program achieves this by allowing the indigenous

inhabitants of these ecosystems to return an essential natural force that has been unnaturally suppressed for decades [43].

**Author Contributions:** Conceptualization, G.R.; methodology, G.R.; formal analysis, G.R.; investigation, G.R.; resources, G.R.; writing—original draft preparation, G.R., J.G.C., D.F., M.M., A.M.H., E.M., P.C.; writing—review and editing, G.R., J.G.C., D.F., M.M., A.M.H., E.M., P.C.; supervision, J.G.C., D.F.; project administration, J.G.C., D.F.; funding acquisition, J.G.C., D.F. All authors have read and agreed to the published version of the manuscript.

**Funding:** This research, along with the APC, was funded by a Joint Venture Agreement (19-JV11221636-135) between the U.S. Forest Service Rocky Mountain Research Station and Colorado State University Department of Journalism and Media Communication.

**Institutional Review Board Statement:** The study was conducted according to the guidelines of the Declaration of Helsinki, and approved by the Institutional Review Board of Colorado State University (protocol code 19-9590H; originally approved on 17 December 2018; updated on 6 December 2019).

**Informed Consent Statement:** Informed consent was obtained from all participants involved in the study.

**Conflicts of Interest:** The authors declare no conflict of interest.

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
