# Peer review of "Doing Work on the Land of Our Ancestors: Reserved Treaty Rights Lands Collaborations in the American Southwest"

_fire, doi:10.3390/fire4010007_

Round 1
Reviewer 1 Report
This paper is certainly a Perspective even if it is not what they thought it was. The paper raises several issues but mostly the issues discussed are those about the bureaucracy i.e., the difficulty of different agencies working together and the use of indigenous firefighters in prescribed burns. Nowhere does the paper suggest solutions and even routes for these solutions. It seems to be primarily a polemic about the difficulties and how there are monies available that are not being used.
I am not sure this journal is the proper venue. It is written as one would expect by a journalist. Most of the paper is about the forest service management goals of fuel reduction and the use of indigenous fire crews in this reduction on trust land.
If I was an indigenous person, I would find this paper offensive. The concern seems to be only that the forest service has tried their best to use indigenous crews and other indigenous elders in the projects. The elders and apparently an indigenous archaeologist were used it seems (this is not clear in the paper) to look for archaeological sites in areas before the burns. Several places they make statements like line 101 that using indigenous elders gave exposure to the forest service of a new ways of looking at the landscape. But nowhere does this new view of the landscape seemed to be articulated in the paper or in the management. Throughout the forest service viewpoint seems to be about the project monies and the indigenous community taking advantage of some of this from the government. Prescribed burning is an agenda of the government often without any real influence on it by the indigenous communities. At no point do we find out what the historical environment that is being restored (except in reference) is and how this relates to the indigenous communities view of both the historic environment and the future environment which they may desire.
Reviewer 2 Report
This research brief highlights findings from interviews with tribal land managers and Forest liaisons in NM and AZ, including the need to heavily weigh tribal input in land restoration projects on federally-controlled ancestral lands and to consider Reserved Treaty Rights Lands (RTRL) funding structures and restrictions in relation to other existing BIA, DOI, and FS structures, reporting requirements, and accomplishment tracking. It also shares the success of the RTRL-financed Pacheco Canyon Prescribed Burn in NM as an example of true interagency collaboration in fire adapted landscape restoration, with equal assessment and inclusion of both cultural and environmental interests.
Interviews—add more descriptive information—were they structured? Open? What did you ask them about? How many interviews? How many sites?--what were the overall themes (could describe in a few sentences) vs. what you highlight in the research brief?
Line 45- research was conducted “over the last year” –specify timeframe so makes sense over time
Reviewer 3 Report
BRIEF SUMMARY
I consider the manuscript represents an interesting, revealing and a valuable effort collecting perceptions of tribal actors (tribal land managers and personnel who work with Tribes) about the Reserved Treaty Rights Lands (RTRL) program and its implementation on tribal and non-tribal national forest lands in the American Southwest. The authors also analyse the effectivity of policies currently implemented by this program on collaborative fuel management projects to prevent and control forest wildfires. Furthermore, the work evaluates the articulation between RTRL program and federal land management agencies (like US Forest Service) linked to fire management, as well as potential gaps in implementing effective and collaborative fire management in the regions studied.
The authors revealed assertively that despite the existence of the RTRL funding mechanism that possesses the capacity to help federal land management agencies, it is underutilized. Through this article, the authors attempt to raise awareness of the crucial importance of activating and supporting the RTRL program to strengthen shared management efforts among tribes and federal land management agencies to meet the goal of restoring fire-adapted ecosystems to historical preconditions.
BROAD COMMENTS
Although this type of publication (perspectives) involves short articles in length and a limited number of references, I recommend including some aspects and clarifications that could improve the scope of this work.
1- The aim of prescribed burns, in general, is to imitate traditional Indigenous burning practices for fuel management purposes. However, it would be useful to describe, even if briefly, what was the wildfire prevention system used by the ancestral tribal communities in the American Southwest.
2- What are the most important tribal groups in this region? It would be advisable to name them. Also, what is the land tenure status vs national parks (or protected areas) figures in the region under consideration?
3- Many keys and dominant institutions of resource and environmental management actually lack the capability to work interculturally, a feature described by Howitt et al. (2013) as “intercultural capacity deficit”. What the authors propose is to reduce the present intercultural capacity deficit of some federal agencies. I recommend to include some references that describe the interculturality as a key, useful and valuable mechanism to implement effective and participatory fire management policies instead of fire suppression in South America, for example:
Bilbao, B.A., et al. (2019) Sharing Multiple Perspectives on Burning: Towards a Participatory and Intercultural Fire Management Policy in Venezuela, Brazil, and Guyana. Fire, 2(3), 39.
Eloy, L. et al. (2018) From fire suppression to fire management: Advances and resistances to changes in fire policy in the savannas of Brazil and Venezuela. J. 185, 10–22, doi:10.1111/geoj.12245.
Howitt, R. et al. (2013) New geographies of coexistence: reconsidering cultural interfaces in resource and environmental governance Asia Pacific Viewpoint 54(2): 123–5.
Mistry, J. et al. (2018) New perspectives in fire management in South American savannas: The importance of intercultural governance. Ambio 48, 172–179, doi:10.1007/s13280-018-1054-7.
4- Is it possible to include some pictures of the meetings or territories mentioned in this paper?
Reviewer 4 Report
I enjoyed reading this informative perspective piece. While I have less experience reviewing "perspective" articles, I note that I needed to read this article a couple of times to clearly understand the main goals. Perhaps I read it first as if reading a research article, none-the-less I think the main message and recommendations can be stressed more. As such I recommend very minor changes.
1) I recommend changing the ABSTRACT. In a research article, the purpose of the abstract is clear, but in a perspective less so, in my view and this one reads as if the authors are attempting to fit a "perspective" into a research abstract format. Why not make it a summary statement. I would argue that the first line of the abstract should change from: "This article explores the Reserved Treaty Rights Lands (RTRL) program and how it has been used in the American Southwest to implement..." to one that is more definitive. Perhaps an intro that ends with the phrase, "...awareness about an underutilized funding mechanism" instead of burying the key finding.
2) I think the article could benefit from a brief summary in the conclusion and Abstract about the key recommendations and findings. The authors speak about the underutilized funding mechanism, but they also raise additional points. For example, I found this quite interesting, "He wants to see the US Forest Service and BIA figure out a way to adjust project accomplishments and reports so that BIA funding opportunities are not dinged for providing RTRL trained crews to do work off reservation lands." Do the authors agree that this should be recommended? Why not conclude with a list of clear recommendations aimed at overcoming the under-utilization issue?
Minor comments:
I would also move the paragraph defining "prescribed burning" down in the text. Most readers of this journal are fully aware of the definition or meaning and it seems out of place in the text.
Can the authors provide a small amount of additional information about the interview format? Was there a list of questions? Informal, Semi-formal?
Ref 13: Pyne is a great resource for general policy and broad fire use changes. Is there a reference here of specific study of the fire regime changes in the local or regional area? Tree-core/fire scar data? I ask because in California, where we work, there is great variation in these issues--some places too little fire, others too much.
Hope I was helpful.
Reviewer 5 Report
This is a timely and compelling perspective article that fits well with the scope of the journal. It is clearly and well written. The methods and incorporation of results are appropriate and strong – they give some legitimacy to tribal perspectives. It is commendable that tribal and Indigenous perspectives have been brought in to the manuscript. However, the article will have much greater cogency and incorporation of Indigeneity if tribal voices are represented via co-authorship of this paper. I strongly recommend that the authors reach out to several of their tribal collaborators to request their review of the paper (it sounds like this may have already occurred) and invite their addition as listed co-authors. While ideally this would occur at an earlier stage in the manuscript preparation it is not too late to take this important step, the article is titled “Working on the lands of OUR ancestors …” after all. It is noted that there is no single tribal or Indigenous voice and that it is probably not realistic to get buy-in from all of the tribal representatives involved, perhaps those from the Pueblo of Santa Ana and Pueblo of Tesuque.
One minor suggestion would be to add a citation/reference or two to work on stakeholder engagement with Indigenous communities in the southwest (e.g., Karletta Chief's work, i.e., Chief et al 2016. Engaging Southwestern Tribes ..." Water) to provide additional context to the approach being used in this paper.
Author Response
Thank you for your positive feedback. It means a lot to us.
Although we do not know your identity, we do know that the Editor in Chief of Fire asked you to review the article because of your previous research experience working with tribal communities. This is the first time the authors (original 3) have worked closely with tribes, so it is reassuring to know that you think we are going about our research appropriately.
And your recommendation to include participants as authors only makes sense for the reasons you provided. This was the missing piece hiding in plain sight. Thank you for revealing it to us. We are happy to report that 4 of the 7 participants have agreed to sign on as co-authors. Given the participatory research context of this study, it really is fitting to list the participants as co-authors, since all 4 helped to compose the paragraphs they appear in. We listed authors 4-7 according to the amount of actual writing they added to the manuscript.
Per your recommendation, we incorporated the Chief (2016) article. It was very illuminating. We think this reference—along with Smith’s (1999) Decolonizing Methodologies—serves to round out our participatory research approach.
Thank you for taking the time to share your insights with us.
Round 2
Reviewer 1 Report
The paper is now turned into a much more important paper. I particularly like the addition of what can be done suggest solution to the policy issues. Thanks for your major effort in rewriting the paper.
Author Response
Thank you for your feedback. I am glad that the paper resonated with you this time around.